# Social Support for People with Morbid Obesity in a Bariatric Surgery Programme: A Qualitative Descriptive Study

**DOI:** 10.3390/ijerph18126530

**Published:** 2021-06-17

**Authors:** María José Torrente-Sánchez, Manuel Ferrer-Márquez, Beatriz Estébanez-Ferrero, María del Mar Jiménez-Lasserrotte, Alicia Ruiz-Muelle, María Isabel Ventura-Miranda, Iria Dobarrio-Sanz, José Granero-Molina

**Affiliations:** 1Hospital HLA Mediterráneo, 04007 Almería, Spain; mjtorrentes@gmail.com (M.J.T.-S.); Manuferrer78@hotmail.com (M.F.-M.); 2Hospital Universitario Torrecárdenas, 04009 Almería, Spain; beatriz.estebanez.sspa@juntadeandalucia.es; 3Department of Nursing, Physiotherapy and Medicine, University of Almeria, 04120 Almería, Spain; mjl095@ual.es (M.d.M.J.-L.); arm297@ual.es (A.R.-M.); mvm737@ual.es (M.I.V.-M.); ids135@ual.es (I.D.-S.); 4Faculty of Health Sciences, Universidad Autónoma de Chile, Santiago 7500000, Chile

**Keywords:** social support, morbid obesity, bariatric surgery, qualitative study

## Abstract

Background—Morbid obesity (MO) is a chronic metabolic disease affecting physical, psychological and social wellbeing. Bariatric surgery is a reliable method for losing weight in the long term, improving the quality of life, body image and social life of people with MO. Current literature recognises the importance of social support in controlling weight and coping with MO. The objective of this study was to describe and understand experiences related to social support for patients with MO included in a bariatric surgery programme. Methods—A qualitative descriptive study, where data collection included thirty-one interviews with people diagnosed with MO involved in a bariatric surgery programme. Results—Three main themes emerged from the analysis: (1) accepting the problem in order to ask for help, (2) the need for close support and (3) professional support: opposing feelings. Conclusions—A partner, family and friends are the key pillars of social support for those with MO included in a bariatric surgery programme. Healthcare professionals gave formal support; the bariatric surgery team provided information, trust and assurance. Nurses provided healthcare 24 h a day, making them the main formal support for people in the bariatric surgery programme.

## 1. Introduction

Obesity is a chronic disease that is characterised by an accumulation of fat or hypertrophy of fatty tissue. Obesity is a public health issue, although the prevalence varies depending on the country [1]. It affects approximately 13% of the global adult population [2]. The prevalence of obesity in Spain is 22.8% of men and 20.5% of women [3]. Obesity is defined as having a body mass index (BMI) > 30 kg/m^2^, and morbid obesity (MO) as having a BMI between 40–49.9 kg/m^2^ [4,5]. Obesity is a multifactorial problem that is associated with metabolic, endocrine [6] and vascular problems, as well as diabetes and hypertension [7]. Added to this are external factors, such as poverty [8], unhealthy diet, lifestyle and physical exercise, as well as socio-cultural factors [9]. MO is a health risk factor, as it is related to anxiety, depression, body image disorders and sexual dysfunction [10,11]. MO can be prevented with a multidisciplinary approach, including health education, diet, physical exercise and pharmacological treatment [12,13,14]. However, therapies or treatments are not always effective and therefore surgical treatment is required [15]. Bariatric surgery is a reliable method for losing weight in the long term, improving the quality of life, body image and social life of people with MO. The criteria for inclusion for a person with MO in a bariatric surgery programme are: to be an adult with a BMI ≥ 40 kg/m^2^ or an adult with a BMI between 35–39.9 kg/m^2^ and a serious associated comorbidity, such as type 2 diabetes, high blood pressure or hyperlipidemia [16]. Bariatric surgery involves a set of surgical procedures to treat obesity, prevent the associated morbidity and mortality and improve the quality of life of obese patients following weight loss. Bariatric surgery includes restriction techniques that reduce gastric volume [17], malabsorptive techniques and mixed techniques [18].

Physical, psychological, social and cultural factors make living with MO difficult. People with MO face physical impediments, difficulty in moving [19], low self-esteem or self-control [20], disorders associated with negative body image [21], shame and psychosocial stress, all of which can lead to limited social relations, stigma and the loss of quality of life [22,23]. Facing MO not only requires weight loss but also dealing with psychological and social problems [24]. Social support refers to the social resources that are available for people facing a health issue, which are provided by formal and/or informal support groups. Social support is key in preventing and overcoming chronic pathologies, such as diabetes, cardiovascular disease, osteoarticular disease or MO [25,26].

Various studies have demonstrated the importance of social support in different phases of obesity [27]. Social support is associated with making the decision to undergo bariatric surgery [28], post-surgery results [29,30,31,32], stress levels [33], information prior to treatment [34] and coping strategies [35]. Other studies have explored the experiences of patients in MO support groups after undergoing bariatric surgery [36,37] or patients who are expecting bariatric surgery alongside their partners [38,39]. Although the literature recognises the importance of social support in controlling weight and overcoming MO, the experiences of people with MO in a bariatric surgery programme have scarcely been studied.

In line with Leahy-Warren’s social support theory [40], social support has a structural and functional dimension. The structural dimension refers to the social networks of people with MO, which are divided into informal (family, friends and acquaintances) and formal (essentially healthcare professionals). The functional dimension is related to the exchange of activities (informative, instrumental and emotional support), types of social support received and the perception/satisfaction of them [41]. The objective of this study was to describe and understand the experiences regarding the social support of people with MO that were included in a bariatric surgery programme.

## 2. Materials and Methods

### 2.1. Design

A qualitative descriptive study was used. This methodology allows for little-known experiences to be described from a naturalist perspective without losing their essence in the analysis process. The study was developed and reported in accordance with the Consolidated Criteria for Reporting Qualitative Research (COREQ) [42].

### 2.2. Participants and Setting

The researchers used purposive sampling to select people with MO that were included in a bariatric surgery programme. The participants were recruited by surgeons and nurses associated with the bariatric surgery programme for people with MO. The criteria for inclusion were: be a person diagnosed with MO at least 10 years prior to the study, in the phase prior to surgical intervention, a Spanish speaker, over 18 and having signed an informed consent form. The exclusion criteria were: rejected participating in the study, not be included in a bariatric surgery programme or be in a different phase of treatment. Out of 44 people with MO selected for telephone contact and recruitment, two did not answer the phone, nine refused to discuss the topic and two did not have time to be interviewed. The interviews took place with a total of 31 participants, of which, 21 were women and 10 were men (Table 1). The study was carried out in two hospitals in the south of Spain between April 2019 and January 2020.

### 2.3. Data Collection

Nurses called participants who fulfilled the inclusion criteria to explain the objectives, respond to queries and request voluntary participation in the study. Nurses and participants had established contact prior to the clinical care context. The nurses had a Master’s degree in qualitative research and they did not need further training. The participants agreed to be interviewed in a hospital meeting room outside of the surgical ward. Two nurses with five years of experience working in the bariatric surgery programme carried out in-depth interviews following the interview protocol (Table 2). Each participant did one individual, private, in-depth interview that lasted 56 min on average. The nurses collected field notes to describe the data. Before the interview, participants were informed about the study, signed consent was obtained and sociodemographic data of the participants were collected. The participants’ answers were recorded, transcribed and analysed using ATLAS.ti.9 (ATLAS.ti Scientific Software Development GmbH, Berlin, Germany). Data collection ceased when the researchers considered that they had reached data saturation.

### 2.4. Data Analysis

Thematic analysis was carried out using Braun and Clarke’s [43] method with the following steps: (1) familiarisation with the data—data transcription, reading and re-reading and annotation of initial ideas; (2) generate initial codes—systematic codification of data groups; (3) find themes—convert codes into themes; (4) revise themes—verify codes with themes; (5) define themes—analyse the details of each theme; (6) elaborate the report—select examples of themes and subthemes, associate the analysis with the research question and generate the final report.

### 2.5. Rigor

Credibility: participants and researchers were familiarised with the context of the study to ensure trust and obtain rich data. Various researchers participated in the codification, analysis and interpretation of the data. Transferability: the researchers described the experiences and context of the study in detail. Reliability and confirmability: the nurses did the transcriptions and then other members of the research group revised them, verified them and accepted the participants. Reflexivity: the researchers examined their own values and preconceptions about the issue at hand before describing the results.

### 2.6. Ethical Issues

The study was carried out in accordance with the ethical standards established by the Helsinki Declaration. It was approved by the Ethical Committee of the University of Almería’s Nursing, Physiotherapy and Medicine Department (protocol number: 45/2018). Prior to the study, informed written consent and permission to record the interview were obtained. The participants were informed that their participation in the study was independent of the bariatric surgery programme.

## 3. Results

A total of 31 patients, men (32.26%) and women (67.74%) with a mean age = 44.35 years (SD = 11.57 years, range = 20–66 years) were recruited. The BMI of the participants was 44.5 ± 6.57 kg/m^2^, MO started during adolescence (32.26%) or in adulthood (32.6%) (Table 1). Three themes allowed for the description and understanding of the social support experienced by patients with MO in a bariatric surgery programme (Table 3).

### 3.1. Accepting the Problem in Order to Ask for Help

Those with MO faced physical, psychological and social problems. Alongside the associated risks and comorbidites, MO was problematic in relation to the patient’s physical appearance, image, emotional state and the demands of daily life. Acknowledging the situation was key in being able to face it, ask for help and find social support.

#### 3.1.1. Social Stigma, the Other Weight of Obesity

Those with MO found themselves on the receiving end of insulting comments, which could lead to feelings of frustration, shame and blame. This situation could happen as early as childhood, resulting in the victims becoming more and more sensitive and insecure. As one participant stated, it was possible to interpret any gesture or look as contemptuous:
*I think they look at me,…I don’t go unnoticed and my mind doesn’t rest. If I eat an ice-cream I think that everyone is thinking “well, that’s why she’s so fat”. I can’t go on like this.*(IDI1)

Social networks, the media and society’s notion of beauty as being thin led to prejudice against people with MO. The interviewees found it difficult to buy clothes, find jobs and establish social relations, both in terms of leisure and finding a partner. The social stigma around MO had more of an impact than physical problems, which was exemplified by the pity people with MO face in relation to their limitations:
*My colleagues are always looking out for me “leave that, I’ll do it,” “careful, you might fall, be careful”. I feel like they are pitying me…I don’t want that!*(IDI24)

Those with MO found that understanding and empathy came from people who were in the same situation. Putting yourself in someone else’s shoes is difficult if you have never experienced the sorts of comments and blaming looks from society when faced with MO.

Those with MO turned to patient associations to talk about their problems and experiences and to find social support.

#### 3.1.2. Acknowledging MO

Obesity is a chronic disease and the majority of those affected took their time to acknowledge the problem. They did not feel capable of overcoming it or asking for help from professionals or their close social circles. Feelings of rejection and denial were very common in the early phases of acknowledgement, at which point, social support played an important role:
*The doctor said that I had to look for solutions but I didn’t want to acknowledge it or look for help…first you have to become aware of the situation yourself.*(IDI23)

Participants suffered from psychological and emotional disorders and despite being able to share the problem or ask for informal help, they did not find it easy. They recognised that they did not like themselves or feel liked by others, which limited their social life and led to them cutting themselves off. As one participant stated, at those times, it was easier to turn to food than to social support:
*I sometimes argued with my husband and said “right, now I’m going to have some chocolate”. Then I regretted it and even made myself sick so that I wouldn’t put on weight.*(IDI1)

They slowly started to lose contact with those in their social circle, they stopped taking part in leisure activities and it was difficult to find a partner. They found excuses not to leave the house, they withdrew into themselves and looked for help within their family:
*I used to go out more but now I don’t feel like it; I avoid going out with my friends and I go to the beach with my family because I feel more comfortable with them.*(IDI8)

MO presented a constant struggle to stay healthy and in shape, which varied depending on gender, according to the participants. Women associated MO with aesthetic problems; they rejected a body with which they did not identify and avoided looking in the mirror. Whilst they may recognise their physical limitations and the risk posed to their health, they placed more importance on the discomfort and shame related to their body image:
*I see myself as fat, deformed, I don’t recognise myself. When I get out of the shower, I look at myself in the mirror and I don’t want to see myself, I’m ashamed of my body.*(IDI10)

The men did not identify with their bodies either, but MO was more of a health issue for them. As well as the physical limitations and comorbidities, the exacerbation of prior conditions and the inability to carry out daily tasks were what worried them the most. As one participant said, acknowledging the gravity of the situation was the first step in asking for help:
*If I were fat but healthy I wouldn’t mind, the problem is that I’m not healthy. I don’t have a complex, I don’t hide myself, but it’s difficult to deal with alone… I need help!*(IDI26)

#### 3.1.3. Bariatric Surgery, a Shared Journey

After failed attempts to lose weight using other methods, opting for surgery and joining a bariatric surgery programme was a time in which those with MO preferred not to be alone. Participants said that bariatric surgery was not part of their original plan due to fear of the intervention, being judged and the cost of the process. In this phase, rather than family or friends, their partner was the key pillar of support when making these decisions:
*My wife would talk to me about obesity and I would change the conversation. She tried to help me understand that we had a serious problem, that we had to go to the doctor and even consider bariatric surgery.*(IDI27)

Those in the bariatric surgery programme were scared and unsure of the process. If patients had a partner, they shared those feelings with the partner; they needed their partner’s positivity to improve their self-acceptance, for the partner to not feel embarrassed by them and for the partner to encourage them to continue with their daily activities. Those who did not have a partner shared their experiences with family and friends. As one participant said, while they were waiting for the intervention, their expectations of the surgery being a success were high. In fact, they expected a radical change in all areas of their life, which could lead to disappointment:
*I want to be thin, positive, focus on the good things. We (partner) talk about it and I tell him that everything will change, we are going to start from scratch, a new body, different sexuality, another life.*(IDI21)

### 3.2. The Need for Close Support

Those with MO needed support from those with whom they live. Diets, commutes, work and leisure all required effort for which social support was needed. However, they were often faced with a lack of understanding; their perceived notion of social stigma surrounding MO caused an internal conflict that led them to ask for help.

#### 3.2.1. The Partner, the Key Support Figure for MO

Those with MO suffered from stress, insecurity and low self-esteem; it was vital for their partners to accept them. Although the participants did not express it, they thought that their partners were not satisfied with them or their physical appearance, which affected their self-esteem and social or sexual relations. As one participant stated, having their partner’s support was the driving force they needed to opt for and face the surgical process together:
*He tells me that he met me like this (obese), that he loves me like this, that he’s not with me for my physical appearance but I know it’s impossible…I’m embarrassed to have sexual relations.*(IDI6)

The participants’ partners worried a great deal about the associated comorbidities. They feared an exacerbation of cardiovascular or respiratory problems (e.g., a heart attack). This led to the partner being in a permanent state of alert and to adapting their lifestyles to the person with MO. Likewise, the participants had a profound sense of guilt; they knew that their partner was looking out for them and they did not want them to suffer:
*She (wife) is scared of something happening to me, of me dying and leaving her alone. I try to not let my obesity affect my loved ones…but it’s difficult.*(IDI29)

However, this type of support was not universal, and some patients’ partners had made derogatory comments. As one man said, this led to a fear of being left, which, in turn, hastened their decision to enter a bariatric surgery programme.


*In summer, I wear more revealing clothes so my body is on show. If your partner looks at you and says “look at their bodies, not like yours” you think that they are going to leave you and you want to have the operation immediately!*
(IDI 31)

#### 3.2.2. Necessary Company, Unpleasant Scrutiny

Family and friends were social support for those with MO. Nonetheless, some participants felt self-conscious, scrutinised and uncomfortable in their own homes. The family was aware of the severe risks involved with MO for the person diagnosed. They monitored them and hid food, making themselves responsible for the situation not worsening, which became even more evident when the person with MO was awaiting bariatric surgery:
*My family says “don’t eat this, go for a walk, you need to endure it…” Since I was put in the bariatric surgery programme, they monitor me and control me all day, it’s stifling.*(IDI14)

This situation provoked more anxiety, despair and avoidance behaviours. On occasion, they resorted to lies, hiding themselves, putting on alarms and waking up in the middle of the night to eat so that they were not judged in a negative way. They spent years feeling worried and monitored by their families. As one participant stated, although they knew it was for their health, over time, they avoided receiving advice from family members:
*When everyone at home talks about my MO, I get sick of it. I get up from the table and leave. Then I think and say…they’re right.*(IDI28)

Participants indicated that the support they received from their inner circle was fundamental in making decisions. The people close to them made them want to fight and persevere whilst waiting for their bariatric surgery. Their friends could be a source of support that was even greater than that of their families because they listened to them without judging them:
*When I’m feeling down, I turn to a friend and tell her my problems and let off steam. My family just repeat the doctor’s advice over and over, they don’t listen to me or understand me.*(IDI6)

#### 3.2.3. Appearing and Sharing on Social Networks

Those with MO participated in associations, which allowed for mutual support, understanding and health education. They shared their experiences, concerns and hopes, enabling them to feel less lonely and increase social contact. As one of the participants said, these associations were an extension of the safe space provided by close family, in which they could share their problems in a social context.


*The patient association brought us together, it was like a big family. We were social and able to be ourselves, without feeling observed or rejected.*
(IDI19)

Losing this meeting point (MO patient associations shut down at a local level) had a negative impact on the participants’ self-esteem. They tried to replace it by using social networks, such as Facebook, online forums and, above all, Whatsapp, to inform themselves about advances, pre-operative experiences, post-operative complications and expectations of the post-surgery period. It was especially mothers with children with MO who turned to these means of communication:
*Now we use Whatsapp, that’s how we talk, laugh and cry. For a mother, sharing fears with someone who is going through the same as you, who is going to understand you, is priceless.*(IDI11)

### 3.3. Professional Support: Opposing Feelings

Healthcare professionals were the key source of formal support for educating patients and finding a solution to their problem. However, bariatric surgery was not the first choice of treatment, nor was it explained in doctor or nurse consultations.

#### 3.3.1. Lack of Professional Support

When patients acknowledged that they could not face MO alone, they looked for healthcare professionals to guide them in their treatment. However, as some participants said, they did not find the support they had expected in visits to the doctor, nurse or dietician. Many healthcare professionals did not show interest or did not have the necessary training to address MO. The patients note that the professionals did not empathise, listen actively or take their problem seriously, leading them to feel disheartened and less motivated:
*During a medical visit, the nurse weighed me on a kitchen scale. I felt awful. Is that why I go to the hospital? To be embarrassed? I stopped going.*(IDI23)

Dieticians insisted that diet alone allows one to achieve the desired weight loss objective, and they did not support the idea of bariatric surgery. General practitioners or nurses did not inform about bariatric surgery in primary care services either. The majority of patients sought consultation after advice from other patients or finding information on the internet:
*A neighbour told me that she knew people who had been operated, that it was a good team. I only want them to explain it to me properly…to lift that burden.*(IDI19)

#### 3.3.2. The Bariatric Nurse Is “Always There”

From the first contact with the bariatric surgery team, their vision of professional support changed radically. As one participant said, they slowly gained the confidence and assurance that they needed, they felt accompanied, listened to, informed and knew that they could count on their professional support:
*Speaking to an expert outside of my family is liberating. I can be honest, express my feelings without anyone judging me…this team is just amazing!*(IDI1)

Along with the social support from family members and close friends, people with MO needed specialised professional support during the surgical process. The participants appreciated the specialist nurse who guided and accompanied them from entering the bariatric surgery programme up until the post-operative phase. They had access to a 24/7 phone service with the bariatric surgery nurse. It allowed them to ask questions, make appointments, book the surgery, inform themselves and share their fears. Family members also contacted the service to ask for information, calm their nerves or talk about their expectations:
*The specialist nurse manages the situation, always picks up the phone at any hour of the day. It makes you feel assured and confident, you know that you always have someone there to inform, help and advise you…someone who listens to your fears and worries.*(IDI7)

## 4. Discussion

The objective of our study was to describe and understand the experiences of social support amongst people with MO included in a bariatric surgery programme. Leahy-Warren’s [40] theoretical framework allowed us to study how formal, informal and structural support influenced their experiences. MO caused physical, psychological and social problems, as well as emotional and body image issues. According to our results, men regarded MO as a health issue that also affected social and amorous relationships [22]. Women, on the other hand, were more worried about the aesthetic facet, feeling stuck in a body that they rejected and did not want to reveal, which, in turn, deteriorated their social relations [21]. MO limited the daily activities of the patients and although they tried to gradually adapt [12], their frustration led them to look for solutions. The participants experienced an initial phase of denial in which they avoided conversations about MO. After failed attempts at losing weight using other methods, their isolation and frustration led them to join a bariatric surgery programme [44]. However, bariatric surgery is not always an option, some patients prefer other treatments to avoid risk [37] or to hide their intentions due to social or economic obstacles [45].

Various studies confirm the positive effects of social support on health and wellbeing during MO [26], highlighting the importance of the partner, family, extended family and professionals. Concurring with Willmer and Salzmann-Erikson [46], people with MO fear bariatric surgery, but they have high expectations for improving their body image, associated conditions and quality of life. The participants believed that losing weight would strengthen their personal identity, social relations and participation in public and professional life [33]. However, their expectations for positive results could lead to disappointment [44].

The role of weight bias in concerns over body image in bariatric surgery patients could jeopardise the results of their treatment. Obese people tend to percieve their weight incorrectly, even after bariatric surgery, patients seem unable to identify changes in their body image after massive weight loss [47]. Social support could positively influence body image perception and interpersonal body comparisons after bariatric surgery, although additional studies should be carried out.

According to Leahy-Warren [40], informal support is key in making the decision to undergo surgery. In this phase, the partner is a fundamental support figure with whom to share worries, experiences and conflicts, thus offering assurance [48]. Support from one’s partner can improve the phase of awaiting surgery [38], especially if both partners are in the bariatric surgery programme [39]. In accordance with our results, some participants were on the receiving end of negative gestures or comments, leading to reduced communication and affection, as well as increased fear of being left [22]. Support from family is key for people with MO [49] and the more support they have, the more likely they are to consider bariatric surgery [27]. Our results reflect the overbearing attitudes of family members, which the people with MO reject, thus leading them to seek support amongst friends [50]. The quality of social support amongst close members of an inner circle can influence weight gain. In fact, strong social support can be a factor in avoiding weight gain, especially in men [51].

In terms of professional support, the participants criticised the information they received on visits to the doctor or nurse, citing a lack of communication, empathy and possibility to be referred to specialists in bariatric surgery. The patients turned to others who had undergone surgery, associations or the internet (web pages, forums, etc.) to inform themselves about the bariatric surgical programme [46]. Their negative opinion changed radically upon contacting the bariatric surgery team [37], as there was an improvement in information, communication, support and quality of life while waiting for bariatric surgery [38].

Social stigma was an added problem for people with MO [21], as social networks and the media promoted an idealised body image. The participants reported a lack of access to quality support for weight control [52], even though they received support from family and professionals every time they used social networks [53]. Social support could be more effective in long-term weight loss but there was a lack of evidence on the subject [24]. It could be important to encourage people with MO to participate in associations to create a shared social identity [36]. Online support groups can be useful for people with MO in a bariatric surgery programme, as they can discuss their problems and share their fears, expectations, challenges and experiences. Healthcare providers must familiarise themselves with the content of these groups and be cautious when recommending them [35]. Participating in online forums and support groups does not predict stress or weight loss [28], but they could increase awareness for people with MO regarding their responsibility in achieving positive outcomes [29]. This social support can be useful for people with MO in a bariatric surgery programme, although it depends on their level of activity. Professionals must improve their understanding and develop and recommend online self-help groups [25]. The use of social networks, private groups or public pages requires careful examination because their effectiveness has not been demonstrated in experimental studies [54]. Together with early support for patients who take part in a bariatric surgery programme, life-long, individualised support may be required to optimise the effects of bariatric surgery. Receiving follow-up visits from a multidisciplinary medical team or joining support groups may be beneficial in counteracting weight gain after bariatric surgery [55].

## 5. Conclusions

Society’s high beauty standards, the media and social networks all contribute to prejudice towards people with MO. Social stigma, together with physical and psychological problems, demonstrate the need for social support. The majority of participants with MO were slow to acknowledge the problem and joining a bariatric surgery programme was neither an easy decision nor an option they wanted to face alone. They needed information, company and shared experiences. Those with MO had high expectations for bariatric surgery, imagining a radical change in their lives, which could lead to disappointment. The partners of people with MO were the key social support figures for them as they needed support to face the surgical process. However, those with MO were sometimes on the receiving end of disrespectful comments, leading to fear of being left by their partners. The partners were in a constant state of alert, worrying about the potential exacerbation of comorbidities, which provoked feelings of blame amongst the people with MO. Family and friends were also members of the participants’ social support groups. Families were aware of the risks of living with MO and therefore monitored and scrutinised the person’s lifestyle. Those with MO felt self-conscious and uncomfortable, thus their friends substituted family as social support. Healthcare professionals were the main source of formal support, but people with MO did not find the support they expected in general practitioners, nurses or dieticians. Some healthcare professionals showed little interest in the issue and lacked specific training to be able to guide the person with MO in making decisions. The majority of people with MO reached the bariatric surgery consultant either through advice from people who had already undergone the surgery or information found on the internet. This situation changed dramatically upon the first contact with the bariatric surgery team as the people with MO felt confident, assured, accompanied, listened to and informed. Those with MO needed a specialised healthcare professional who centralised the information and accompanied them throughout the whole surgical process. The nurse was the main source of support for those with MO in the bariatric surgery programme; they had access to a 24/7 phone service in which the bariatric surgery nurse attended to their needs. Our study pointed to implications for clinical practice, as well as new research hypotheses. Improvements in perceived social support may positively influence patients in their decision to undergo bariatric surgery. Organising support groups among patients themselves, or with family or friends, managed by bariatric surgery nurses, could help to improve shortcomings and support decision making. Social support can also help to generate more realistic perspectives regarding outcomes after bariatric surgery in patients. The role of healthcare professionals, friends, partners and/or social networks as a means of support for weight control after bariatric surgery may be fundamental and should be investigated further. Providing images of before and after surgery, as well as explaining that the results are not the same in all patients, could be helpful to patients’ perceptions of their improvements.

## Figures and Tables

**Table 1 ijerph-18-06530-t001:** Sociodemographic data.

Participant	Sex	Onset of MO	Marital Status	Job
IDI1	F	Adulthood	Married	Housewife
IDI2	F	Adolescence	Married	Farmer
IDI3	F	Adulthood	Relationship	Housewife
IDI4	F	Adolescence	Single	Unemployed
IDI5	F	Adulthood	Separated	Seamstress
IDI6	F	Pregnancy	Separated	Chef
IDI7	F	Adolescence	Relationship	Administrative assistant
IDI8	F	Adulthood	Married	Healthcare assistant
IDI9	F	Pregnancy	Married	Administrative assistant
IDI10	F	Pregnancy	Married	Farmer
IDI11	F	Pregnancy	Married	Teacher
IDI12	F	Childhood	Married	Administrative assistant
IDI13	F	Adolescence	Married	Housewife
IDI14	F	Adolescence	Single	Unemployed
IDI15	F	Adolescence	Single	Shop assistant
IDI16	F	Pregnancy	Married	Teacher
IDI17	F	Adolescence	Single	Civil servant
IDI18	F	Childhood	Single	Student
IDI19	F	Adolescence	Married	Teacher
IDI20	F	Childhood	Single	Beautician
IDI21	F	Childhood	Relationship	Waitress
IDI22	F	Adolescence	Single	Unemployed
IDI23	M	Adolescence	Married	Engineer
IDI24	M	Adulthood	Married	Salesperson
IDI25	M	Childhood	Single	Farmer
IDI26	M	Adulthood	Married	Salesperson
IDI27	M	Adulthood	Married	Petrol station worker
IDI28	M	Adulthood	Married	Hospitality worker
IDI29	M	Adulthood	Married	Farmer
IDI30	M	Adulthood	Married	Farmer
IDI31	M	Childhood	Married	Unemployed

MO: morbid obesity. IDI: in-depth interview.

**Table 2 ijerph-18-06530-t002:** Interview protocol.

Stage	Subject	Content/Example Questions
Introduction	Motives, reasons	Learn about your experiences of social support since your inclusion in a bariatric surgery programme.
	Ethical issues	Inform about voluntary participation, recording, consent, possibility of withdrawing and confidentiality.
Beginning	Introductory question	Tell me about your experience with MO.
Development	Conversation guide	How has it affected your relationship with your friends and other social relations?
		How has MO affected your perception of your body and your hobbies/free time?
		How has MO affected your life with your partner?
		How has MO affected your relationship with healthcare providers? What kind of formal and informal social support was missing for you after being included in a bariatric surgery programme? What role did technology play in terms of social support in a bariatric surgical programme?
Closing	Final question	Is there anything else you would like to tell me?
	Appreciation	Thank them for their participation, remind them that their testimony will be useful and place ourselves at their disposition.

MO: morbid obesity.

**Table 3 ijerph-18-06530-t003:** Themes, subthemes and units of meaning.

Themes	Subthemes	Units of Meaning
Accepting the problem in order to ask for help	Social stigma, the other weight of obesity	Bullying during childhood, limitations imposed by others, everything is in my head, a look is worth more than a thousand words, understanding is better amongst equals.
	Acknowledging MO	I do not identify with my body, my health comes first, physical limitations, acknowledging, self-punishment, food as comfort and social withdrawal.
	Bariatric surgery, a shared journey	Need to speak, fear, uncertainty, my only hope, sharing experiences, sharing decisions and high expectations.
The need for close support	The partner, the key support figure for MO	Emotional support, fear of being left, lack of communication, suspicion, fear of bariatric surgery and fear of complications.
	Necessary company, unpleasant scrutiny	Feeling observed, discomfort, sharing experiences, feeling scrutinised, feeling heard, a link to reality, selfless help and concern for others.
	Appearing and sharing on social networks	Looking for information, inspecting results and Whatsapp groups.
Professional support: opposing feelings	Lack of professional support	Professional indifference, lack of information, lack of empathy and bariatric surgery is not offered
	The bariatric nurse “is always there”	Trust, assurance, active listening, bariatric surgery team, nurse as a point of reference, 24/7 telephone contact and someone in whom to seek comfort.

MO: morbid obesity.

## Data Availability

Data are available from the authors (J.G.-M., M.J.T.-S. and M.F.-M.) upon reasonable request.

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
