# Peer review of "Social Support for People with Morbid Obesity in a Bariatric Surgery Programme: A Qualitative Descriptive Study"

_ijerph, 2021, doi:10.3390/ijerph18126530_

Round 1

Reviewer 1 Report

The paper is about the qualitative analysis of the social support integrated into service for patients with severe obesity. The study covers an interesting field of research and describes an interesting aspect that usually is taken into low consideration. The paper is well written even if I would suggest to use a more inclusive language without the terminology "morbid", but this is not a rule for this journal. 

I have only few comments for the authors:

- In the introduction, I would focus more on bariatric surgery patients - that is your sample.

- A recent paper (Meneguzzo et al., 2021 - https://doi.org/10.1007/s11695-020-05166-z) has pointed out the role of weight bias in the body image concerns of bariatric patients as a possible factor that could compromise the outcome of the treatments. Social support could help in the modification of the judgment style used by bariatric patients driven by interpersonal body comparisons? Does your service cover this aspect? I think this paper could corroborate what you write about body concerns and drive to thinness.

- I think the paper could include "future clinical steps" that would help the authors to focus their paper from a research and clinical point of view.

Author Response

REVIEWER-1
The paper is about the qualitative analysis of the social support integrated into service for patients with severe obesity. The study covers an interesting field of research and describes an interesting aspect that usually is taken into low consideration. The paper is well written even if I would suggest to use a more inclusive language without the terminology "morbid", but this is not a rule for this journal. 
I have only few comments for the authors:
RESPONSE
The authors appreciate the comments made by Reviewer 1. After consulting the bibliography, we have determined that the term “morbid obesity” (MO) is commonly used in international medical literature for a body mass index between 40-49.9 kg/m2. Thus, we have opted to maintain this term throughout the article.
In the introduction, I would focus more on bariatric surgery patients - that is your sample.
RESPONSE
We agree with the reviewer. The following paragraph has been added to the introduction section:
Bariatric surgery involves a set of surgical procedures to treat obesity, prevent associated morbidity and mortality, and improve the quality of life of obese patients following weight loss. Bariatric surgery includes restriction techniques that reduce gastric volume [17]; malabsorptive techniques and mixed techniques [18].
A recent paper (Meneguzzo et al., 2021 - https://doi.org/10.1007/s11695-020-05166-z) has pointed out the role of weight bias in the body image concerns of bariatric patients as a possible factor that could compromise the outcome of the treatments. Social support could help in the modification of the judgment style used by bariatric patients driven by interpersonal body comparisons? Does your service cover this aspect? I think this paper could corroborate what you write about body concerns and drive to thinness.
RESPONSE
Following the reviewer’s suggestions, we have included a paragraph about the role of weight loss bias in the body image concerns of bariatric patients, referencing the article by Meneguzzo et al. (2021). Improvements in social support could positively influence body image perception and interpersonal body comparisons after bariatric surgery.
I think the paper could include "future clinical steps" that would help the authors to focus their paper from a research and clinical point of view.
RESPONSE
At the end of the conclusions section, implications for clinical practice and new lines of research have been added. 

Reviewer 2 Report

Whilst I have found the manuscript an interesting read, I do have some concerns, particularly about whether Table 1 and the interview excerpts together reveal excess potentially identifiable personal information.

  1. The authors may want to consider if the information provided in Table 1, in conjunction with the interview excerpts may potentially lead to participants and their comments being potentially identifiable, and thus breaching patient confidentiality. Could the precise age, and BMI of participants be replaced by categorial ranges in Table 1. Could this be replaced by a summary statistics table?
  2. How was the interview protocol decided? A lot of the interview questions appear to be focused on MO, and not specifically on their specific experience of bariatric surgery. Did the interview examine whether perceived social/ healthcare support play a role in the participants’ decision to coming forward for bariatric surgery?
  3. Were all the participants still waiting to have their bariatric surgery, or were operations already performed in the study participants, at the time of their study interviews?
  4. Are there any data to indicate if social support (professional support experience, partner support or social network) plays a role in determining long-term weight control in study participants post bariatric surgery?
  5. Have the authors found any evidence that the needs and social support provided differ between the younger participants (e.g. < 35) and the older participants (>45).
  6. A key citation appears to be missing. Tolvanen, L., Svensson, Å., Hemmingsson, E. et al. Perceived and Preferred Social Support in Patients Experiencing Weight Regain After Bariatric Surgery—a Qualitative Study. OBES SURG 31, 1256–1264 (2021). https://doi.org/10.1007/s11695-020-05128-5. This should be discussed in the context of the study findings.
  7. Table 1: Should it be “separated”, rather than “separada”?
  8. Line 40: What is “educationon health”? Please double check.

Author Response

REVIEWER-2
Whilst I have found the manuscript an interesting read, I do have some concerns, particularly about whether Table 1 and the interview excerpts together reveal excess potentially identifiable personal information.
RESPONSE
The authors appreciate Reviewer 2’s comments.
The authors may want to consider if the information provided in Table 1, in conjunction with the interview excerpts may potentially lead to participants and their comments being potentially identifiable, and thus breaching patient confidentiality. Could the precise age, and BMI of participants be replaced by categorial ranges in Table 1. Could this be replaced by  summary statistics table?
RESPONSE
Following the reviewer's instructions, we have eliminated age and body mass index (BMI) data from Table 1. These statistics are summarised at the beginning of the Results section. Confidentiality is thus ensured by avoiding ethical problems.
How was the interview protocol decided? A lot of the interview questions appear to be focused on MO, and not specifically on their specific experience of bariatric surgery. Did the interview examine whether perceived social/ healthcare support play a role in the participants’ decision to coming forward for bariatric surgery?

RESPONSE
â—Ź    Supported by the vast experience of the lead researcher and members of the bariatric surgery team in the treatment of these patients, after extensive bibliographic review and discussion, an agreement on interview protocol was reached.
â—Ź    The interview explored various aspects related to MO in the bariatric surgery programme. Questions were asked about relationships with friends, partners, professionals, etc. Two specific questions about social support were added.
â—Ź    The interview did not specifically explore whether perceived social support from professionals influenced patients' decisions to undergo bariatric surgery, though our results point to this possibility. We have added it to the implications for clinical practice and research at the end of the Conclusions section.

Were all the participants still waiting to have their bariatric surgery, or were operations already performed in the study participants, at the time of their study interviews?

RESPONSE
â—Ź    All of the participants in the study were included in a bariatric surgery programme, and all were awaiting bariatric surgery. 

Are there any data to indicate if social support (professional support experience, partner support or social network) plays a role in determining long-term weight control in study participants post bariatric surgery?

RESPONSE
â—Ź    We cannot draw that conclusion from our results, although this research question allows us to propose a hypothesis to be studied in subsequent research.

Have the authors found any evidence that the needs and social support provided differ between the younger participants (e.g. < 35) and the older participants (>45).

RESPONSE
â—Ź    While patients < 35 years of age seem to be supported more by friends (family members tend to be afraid to recommend bariatric surgery), we cannot draw that conclusion from our results. This is an interesting hypothesis for future studies.

6. A key citation appears to be missing. Tolvanen, L., Svensson, Å., Hemmingsson, E. et al. Perceived and Preferred Social Support in Patients Experiencing Weight Regain After Bariatric Surgery—a Qualitative Study. OBES SURG 31, 1256–1264 (2021). https://doi.org/10.1007/s11695-020-05128-5. This should be discussed in the context of the study findings.

RESPONSE
â—Ź    This is an important citation. We thank the reviewer for their contribution and we have added it to the discussion and the bibliography. 

Table 1: Should it be “separated”, rather than “separada”?

RESPONSE
â—Ź    Changed.

8. Line 40: What is “educationon health”? Please double check.

RESPONSE
â—Ź    Changed.
